# Agroforestry adoption and its influence on soil quality under smallholder maize production systems in western Kenya

**Henry Tamba Nyuma**[1,2]*, **Ruth Njoroge**[1], **Abigael Nekesa Otinga**[1]

**1** Department of Soil Science, School of Agriculture & Biotechnology, University of Eldoret, Eldoret, Kenya,
**2** Department of Agronomy, College of Agriculture & Forestry, University of Liberia, Monrovia, Liberia

* tnyuma@gmail.com

## Abstract

Agroforestry, a sustainable land use practice, was introduced in western Kenya in the early 1990s as a soil fertility replenishment strategy. Since then, the effect of the practice on soil quality has seldom been evidenced. A study was conducted to assess the impact of agroforestry adoption on soil quality under smallholder maize systems in the region. Soil samples were collected from two land use practices: agroforestry adoption (90) and non-agroforestry adoption (30) at 0–30 cm depth from two locations (Busia and Kakamega counties). Results showed variations in soil characteristics between the study locations. Soils in Kakamega contained higher concentrations of soil organic carbon (SOC), exchangeable cations, and micronutrients. On average, agroforestry adoption significantly ($P \leq 0.05$) improved soil physicochemical properties. Under agroforestry adoption, bulk density (BD) reduced by 21% (from 1.4 to 1.1 g cm$^{-3}$), while (SOC) increased by 75% (0.8–1.4%), P by 80% (3.0–5.4 mg kg$^{-1}$), exchangeable K by 256% (0.3–8.0 cmol kg$^{-1}$), Ca by 100% (1.0–2.0 cmol kg$^{-1}$), S by 50% (0.2–0.3 mg kg$^{-1}$), and Cu by 18% (2.8–3.3 mg kg$^{-1}$). Agroforestry adoption significantly increased K and Cu levels above the critical thresholds of 0.4 cmol kg$^{-1}$ and 1.0 mg kg$^{-1}$, respectively, at both locations. *Sesbania sesban* and *Leucaena leucocephala* influenced soil BD, pH and P (4.3.-7.0 mg kg$^{-1}$), exchangeable K (0.4–0.7 cmol kg$^{-1}$), Mg (0.1–0.2 cmol kg$^{-1}$), and Mn (13.5–25.2 mg kg$^{-1}$) at both locations, while *Calliandra calothyrsus* significantly increased SOC in Kakamega only. These findings highlight the significance of agroforestry in soil fertility management. Furthermore, *Sesbania* proved to be effective in enhancing the characteristics of soils at both sites, hence, the need for its inclusion in agroforestry extension messages. Further studies are needed to understand nutrient release mechanisms from agroforestry biomass and their influence on soil characteristics and maize yield in smallholder farming systems.

## 1. Introduction

Healthy soils are cardinal to sustainable agricultural production globally. Soil degradation, as expressed by soil fertility decline, presents numerous challenges to global food and nutrition security. On a global scale, soil degradation affects 3.6–7.5 billion hectares of agricultural

**Data availability statement:** All relevant data are within the manuscript and its Supporting Information files.

**Funding:** This research was funded with supports from the University of Liberia, through the Department of Institutional Development and the Regional Universities Forum for Capacity Building in Agriculture (RUFORUM) under the Graduate Teaching Assistantship (GTA) Grant (Grant No. RU/2024/GTA/CCNY/15), awarded to HTN. The funders had no role in study design, data collection and analysis, decision to publish, or manuscript preparation.

**Competing interests:** Enter: The authors have declared that no competing interests exist.

soils annually, hence threatening global food supply [1,2]. According to the United Nations, land degradation affects about 40% of the planet's land, directly affects half of humanity, and threatens roughly half of the global GDP (US$44 trillion) [2]. Soil degradation is linked to inappropriate agronomic practices, resulting in detrimental effects on soil ecosystem services, including nutrient availability and carbon sequestration [3–5]. Globally, studies have reported the effects of agroforestry on agroecosystems across different ecological zones. These include enhancement of soil biodiversity, carbon sequestration, soil quality, and biomass yield, hence sustainable production [6,7]. Soil quality is an important indicator of agricultural productivity, defined as the ability of the soil to sustain the productivity, diversity, and environmental services of terrestrial ecosystems [8]. The soil's physical, chemical, and biological properties and climatic factors such as temperature and rainfall influence soil quality. Soil physicochemical characteristics such as bulk density, organic carbon, macronutrients (N, P, S), cations (K, Ca, Mg), and micronutrients are reported to significantly influence soil moisture, soil microclimate, microbial communities, and nutrient cycling, hence food and nutrition security, especially in tropical regions, according to Bhardwaj *et al.* [9], agroforestry species such as *Robinia pseudoacacia* and various fruit tree species, like apple, plum, apricot, and peach, are the main agroforestry tree species in the temperate parts of the world. These species possess the potential for carbon sequestration, soil microclimate modification, and ecosystem service. In tropical regions such as Africa, agroforestry is the source of fodder, timber, shade, and a sustainable soil conservation practice, given the extent of soil fertility decline on the continent [10,11]. The impacts of soil degradation on the physical, chemical, and biological functions of soils have been reported across different regions and agroecological zones on the continent, hence the need for a sustainable approach to overcome the challenges associated with soil fertility decline.

In Sub-Saharan Africa (SSA), soil degradation affects more than 40 million hectares of arable lands and the livelihood of more than 120 million farming households. In Kenya, soil degradation affects more than 30% of the country's arable land [12,13] with observed deficiencies of nitrogen (N), phosphorus (P), zinc (Zn) and organic carbon [14]. Maize is an important staple food crop for over 75% of Kenya's population. Its production by smallholder agriculture systems is characterized by soil fertility decline, intensive cultivation, and low productivity [15]. Climate variability and unsustainable land use practices are linked to low productivity of maize, accounting for 30–60% of maize yields in SSA [7]. The situation is further aggravated by soil nutrient imbalances, resulting in the deficiency and or toxicity of essential macro and micronutrients [16–18], which affects agricultural productivity and the overall nutritional security of the country [19,20]. Practically, access to and the affordability of inorganic inputs are among the constraints affecting agricultural productivity in most parts of the country [21]. Alternative soil replenishment strategies are inevitable for sustainable agricultural productivity in the country. Agroforestry is a sustainable land use practice that contributes to sustainable food production and the amelioration of environmental quality through the enhancement of soil ecosystem functions [22–24]. The adoption of fast-growing agroforestry trees such as *Calliandra calothyrsus*, *Gliricidia sepium*, *Leucaena leucocephala*, and *Sesbania sesban,* commonly known as fertilizer trees, are noted for their multiple functions, such as the provision of fodder [22]; biofuel [25,26], soil conservation and carbon sequestration [3–5]. Studies by [27,28] demonstrated the ecosystem benefits of these leguminous species as organic amendments due to the rapid decomposition and nutrient release from their biomass. Gupta et al. [29] reported the amelioration of ecosystem functions and the restoration of degraded soils under agroforestry across different agroecological zones in tropical regions. In Kenya, agroforestry has been documented to contribute to the diversification of farm outputs, climate change mitigation, and the restoration of

degraded soils [30,31]. However, its adoption for soil fertility management and specifically for improved yields of food crops in the country is low, usually attributed to reduced land size, limited knowledge of the management of exotic agroforestry species, and ineffective extension services [11,32]. Reducing knowledge gaps in soil fertility management through agroforestry interventions has the potential to rejuvenate degraded agricultural land and improve household income and food security. Agroforestry trees were planted in western Kenya in the 90s during the campaigns for agroforestry by Kenya Forest Research (KEFRI). Since then, few or no studies have been carried out to establish their long-term effect on physiochemical properties. This study attempts to demonstrate the effects of selected agro-forestry tree species on the physiochemical properties of soils concerning maize production in western Kenya. The results herein can be applied in areas where a similar establishment of AF tree species was introduced on the continent around the same time. Therefore, the study aimed to evaluate the effects of agroforestry adoption on soil quality under smallholder maize production systems in western Kenya. It was hypothesized that agroforestry adoption enhances soil quality for optimum maize production.

## 2. Methods

### 2.1 Study area and site selection

The study was conducted from January to July 2023 in Butsotso ward, Lurambi Sub County of Kakamega County, located at longitude 0°17'3.19"N and latitude 34°45'8.24" E and in Amukura, Chakol North Ward, Teso South Sub County of Busia County located at longitude 0°25'59.99" N, latitude 34°08'60.00" E). Both study locations are situated in the Midland (LM) Zone 2–3 and Upper Midland (UM) Zone 4 Agro-ecological zones in western Kenya [33] (Fig 1).

Rainfall in western Kenya is above average, showing a bimodal distribution with two distinct wet seasons. The long rains occur from March to May, and the short rains from

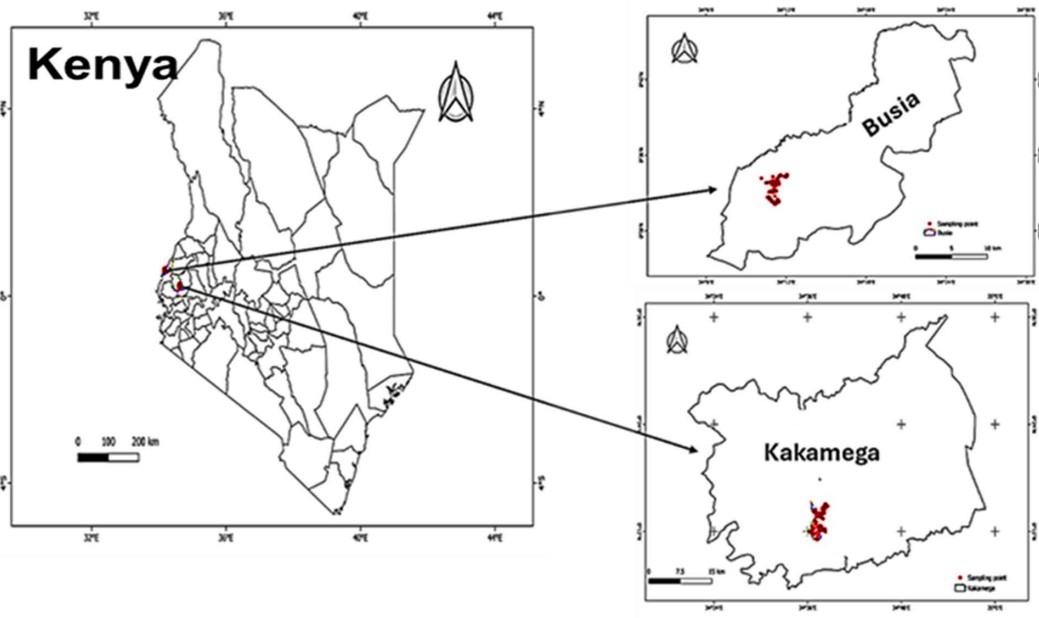

**Fig 1. The map of Kenya showing the study locations and sampling points (red dots).**

October to December. Average rainfall during long rains is 1000–1200 mm, while short rains are 500–800 mm annually [31]. Soils in the study area are highly weathered and acidic with low inherent soil fertility status (low N and P, as well as Ca, Mg and K). The study sites were chosen based on the status of soil fertility decline in the region and the lack of data on the long-term impact of agroforestry adoption on smallholder maize cultivation systems since the introduction of agroforestry in western Kenya. The dominant soil types in the study area are broadly classified as Ferralsols and Nitisols in Busia and Kakamega, respectively [34,35].

## 2.2  Study design

The study sites were purposely selected based on similarity in agro-climatic conditions and the adoption of agroforestry activities [36]. The study involved a field survey using an open-ended questionnaire per household and collection of soil samples for laboratory analyses to assess the effects of adoption practices (agroforestry adoption and non-adoption), and selected agroforestry tree species Calliandra (*Calliandra calothyrsus*) Leucaena (*Leucaena leucocephala*) and Sesbania (*Sesbania sesban* (L.) Merr.) on soil quality for maize production.

## 2.3  Soil sampling

A total of 120 soil samples were collected from 60 locations at 0–30 cm depth in Busia and Kakamega counties, based on agricultural extension officers' recommendations. An uneven sample size (90 agroforestry adopters and 30 non-adopters) was drawn given the dominance of agroforestry practices in the study communities, following sampling guidelines prescribed by [37]. Soil samples under agroforestry adoption were randomly collected at 0–30 cm depth within a 1 m radius from the base of agroforestry trees in three directions and properly mixed to form composite samples. Sampling in non-agroforestry adoption plots was done randomly from farmers' fields; however, in fields where crops were established, soil samples were collected between crop rows to ensure minimum interference with their roots. All samples were properly labelled, packaged in khaki paper bags and transported to the Department of Soil Science Laboratory at the University of Eldoret (111–115 km) away from the study sites for physicochemical analyses.

## 2.4  Field and laboratory soil measurements

All field sampling and laboratory analytical procedures were conducted following protocols described by [37]. Particle size distribution was determined by the Bouyoucos hydrometer [38], while soil bulk density (BD) was determined in situ by the core ring method [39]. Soil pH was determined in 1:2.5 $H_2O$ suspension using a glass electrode pH meter (model: HI 2211, Hanna instruments). The Kjeldahl method [40] and the potassium dichromate method [41] were based on the total soil N and soil organic carbon (SOC) of the soil. Available P was determined following the Olsen method. The exchangeable cations, Ca (cmol kg$^{-1}$) and Mg (cmol kg-1), were examined by atomic absorption spectroscopy, while K (cmol kg-1) was analyzed by flame photometry. Soil micronutrients, viz Cu (ppm), Fe (ppm), Mn (cmol kg$^{-1}$) and Zn (ppm) were determined by atomic absorption spectroscopy after ethylenediaminetetraacetic acid (EDTA) extractions as described in (Table 1).

## 2.5  Data analysis

Data was organized in Microsoft Excel (2013 version) and subjected to statistical analysis using R version 4.2.2 [49]. Significant differences ($p < 0.05$) in soil properties were assessed using linear mixed-effects models (lme4 package). Least significant difference (LSD) pairwise post hoc tests were performed for means comparison at $p < 0.05$, where factors were identified as statistically

**Table 1. Soil analysis, nutrient threshold levels Hazelton& Murphy [42], and critical values for maize growth in Kenya as recommended by NAAIAP& KARI [43].**

| Soil parameter | Assessment method* | General Interpretation** | | | | | Interpretation for Kenya | |
|---|---|---|---|---|---|---|---|---|
| | | Very low | Low | Moderate | High | Very high | Critical range | References*** |
| Soil texture (%) | Bouyoucos hydrometer method | na | na | na | na | na | na | [38] |
| Bulk Density (g cm⁻³) | Core ring | <1.0 | 1.0–1.3 | 1.3–1.6 | 1.6–1.9 | >1.9 | na | [39] |
| SOC (%) | Potassium Dichromate | 0.04–0.06 | 0.06–1.00 | 1.00–1.80 | 1.80–3.00 | >3.00 | <2.7 | [41] |
| pH | 1.1:2.5 H$_2$O | 5.1–6.0 | 6.1–6.5 | 6.6–7.3 | 7.4–8.4 | >8.5 | <5.5 | [44] |
| Total N (%) | Acid digestion | <0.05 | 0.05–0.15 | 0.15–0.25 | 0.25–0.50 | >0.5 | <0.2 | [40] |
| P (mg kg⁻¹) | Olsen | <5 | 5–10 | 10–17 | 17–25 | >25 | <30.0 | [45] |
| K (cmol kg⁻¹) | Ammonium acetate | 0–0.2 | 0.2–0.3 | 0.3–0.7 | 0.7–2.0 | >2 | <0.4 | [46] |
| Ca (cmol kg⁻¹) | Ammonium acetate | 0–2 | 2–5 | 5–10 | 10–20 | >20 | <2.0 | [46] |
| Mg (cmol kg⁻¹) | Ammonium acetate | 0–0.3.0 | 0.3–1 | 1–3 | 3-8 | >8 | <1.0 | [46] |
| S (mg kg⁻¹) | Turbidity method | na | na | na | na | na | <5.0 | [47] |
| Cu (ppm) | EDTA | na | na | na | na | na | <1.0 | [48] |
| Fe (ppm) | EDTA | na | na | na | na | na | <10.0 | [48] |
| Mn (ppm) | EDTA | na | na | na | na | na | <0.1 | [48] |
| Zn (ppm) | EDTA | na | na | na | na | na | <5.0 | [48] |

*The method of analysis prescribed by different authors compiled in Okalebo Manual was adopted by the Department of Soil Science, University of Eldoret.

**Soil threshold levels and interpretation Hazelton& Murphy 2007 [42].

***Critical values are based on recommendations NAAIAP& KARI 2004 [43].

na = values not available (i.e., not postulated) in the cited literature.

BD, bulk density; SOC, Soil organic carbon; TN, Total Nitrogen; P, available/Olsen phosphorous; K, Potassium; Ca, Calcium; Mg, Magnesium; S, available sulphur; Mn, Manganese; Fe, Iron; Zn, Zinc; Cu, Copper.

significant for adoption practices and study locations. A one-way ANOVA was conducted to determine the effects of agroforestry tree species on soil characteristics, and means were declared significant at ($p < 0.05$) according to Tukey's Honesty Test. Soil quality indicators (BD, SOC pH, exchangeable bases, and macro and micronutrients) were assessed as prescribed by Okalebo et al. [37]. Thereafter, the results of soil physicochemical characteristics were interpreted using thresholds previously compiled and summarized by Hazelton & Murphy [42] and compared against the critical nutrient levels for maize growth in Kenya, as recommended by NAAIAP& KARI [43].

## 3. Results

### 3.1 Influence of agroforestry adoption and study location on soil characteristics

Findings from the study showed significant differences in soil physicochemical properties between adoption practices and study locations (Table 2). Between the study locations, the results showed that soils in Kakamega had higher clay (19.2%), SOC (1.5%), TN (0.2%), K (0.8 cmol kg⁻¹), Ca (2.2 cmol kg⁻¹), Mg (0.4 cmol kg⁻¹), Cu (3.7 mg kg⁻¹), Fe (3.7 mg kg⁻¹), and Mn(27.1 mg kg⁻¹) than soils in Busia (Table 2). The mean BD in Busia (1.3 g cm⁻³) was significantly ($P \le 0.05$) higher than that of Kakamega (1.2 g cm⁻³). However, a slight change in BD (1.3–1.25 cm⁻³) at both locations indicates the potential of agroforestry adoption to reduce BD (Table 2). Soil organic carbon differed significantly ($P \le 0.05$) between the study locations and between adoption practices (Table 2). Soil organic carbon ranged from 0.1% and 3.6%, with a mean of 1.0%) in Busia and 1.5%) in Kakamega. Significant ($P \le 0.05$) increases in SOC under agroforestry adoption 1.0–1.1% in Busia and 0.9–1.7% in Kakamega as observed

**Table 2. Mean physico-chemical characteristics of soils as influenced by agroforestry adoption and study location.**

| Soil properties | Busia | | | Kakamega | | |
|---|---|---|---|---|---|---|
| | Adopters | Non adopters | Mean Busia | Adopters | Non adopters | Mean Kakamega |
| BD (g cm$^{-3}$) | 1.2(±0.1)b | 1.3(±0.1)a | 1.3(±0.1) | 1.1(±0.1)a | 1.3(±0.1)b | 1.2(±0.1) |
| SOC (%) | 1.1(±0.1)a | 0.7(±0.1)a | 1.0(±0.1) | 2.0(±0.1)a | 1.0(±0.2)b | 1.5(±0.1) |
| pH (H$_2$O) | 5.8(±0.1)b | 6.3(±0.1)a | 5.8(±0.1) | 5.5(±0.1)a | 5.7(±0.1)a | 5.5(±0.1) |
| TN (%) | 0.2(±0.1)b | 0.3(±0.1)a | 0.1(±0.1) | 0.2(±0.1)b | 0.4(±0.1)a | 0.2(±0.1) |
| P (mg kg$^{-1}$) | 5.5(±0.3)a | 4.3(±0.5)a | 5.2(±0.1) | 5.3(±0.2)a | 1.6(±0.5)b | 4.3(±0.3) |
| K (cmol kg$^{-1}$) | 0.5(±0.1)a | 0.4(±0.1)b | 0.5(±0.1) | 1.0(±0.1)a | 0.3(±0.1)b | 0.8(±0.1) |
| Ca (cmol kg$^{-1}$) | 1.0(±0.1)a | 0.8(±0.1)b | 1.0(±0.1) | 3.0(±0.2)a | 1.1(±0.3)b | 2.2(±0.1) |
| Mg (cmol kg$^{-1}$) | 0.2(±0.1)a | 0.1(±0.1)a | 0.1(±0.1) | 0.4(±0.1)a | 0.3(±0.1)a | 0.4(±0.1) |
| S (mg kg$^{-1}$) | 0.2(±0.1)a | 0.2(±0.1)a | 0.3(±0.1) | 0.3(±0.1)a | 0.2(±0.1)a | 0.2(±0.1) |
| Cu (ppm) | 2.7(±0.1)a | 2.3(±0.1)b | 2.6(±0.1) | 3.9(±0.1)a | 3.3(±0.1)b | 3.7(±0.1) |
| Fe (ppm) | 2.5(±0.1)a | 2.4(±0.1)a | 2.5(±0.1) | 3.7(±0.1)a | 3.8(±0.3)a | 3.7(±0.1) |
| Mn (ppm) | 22.0(±0.5)a | 14.0(±1.0)b | 20.0(±1.0) | 29.0(±0.8)a | 23.1(±1.2)b | 27.1(±1.0) |
| Zn (ppm) | 0.9(±0.1)b | 1.2(±0.1)a | 1.0(±0.1) | 1.6(±0.2)a | 1.1(±0.3)a | 1.4(±0.1) |

Means with different letters within a row indicate significant differences (p ≤ 0.05) for columns representing different factors (i.e., agroforestry adoption practice and adoption * location).

represented a 5% and 49% increase in SOC in Busia and Kakamega, respectively (Table 2). Soil pH recorded in the study ranged from 4.5–6.9, and in the range of strongly acidic (4.5–5.5) to slightly acidic (5.6–6.5) according to (39). Differences in soil pH were observed at both locations and adoption practices (Table 2). Farmers adopting AF showed a significant reduction in pH from 6.3 to 5.5 at both locations (Table 2). Within the study locations, it was revealed that agroforestry adoption significantly reduced BD (1.3–1.2 g cm$^{-3}$), and TN (0.2–0.3%), but increased SOC (0.7–1.1%), K (0.4–0.5 cmol kg$^{-1}$), Ca (0.8–1.0 cmol kg$^{-1}$), Cu (2.3–2.7 ppm), and Mn (14.0–22.0 ppm). However, pH, TN, and Zn decreased significantly under agroforestry adoption in Busia, but P, Mg, S and Fe did not differ statistically between adoption practices in Busia. Similar patterns were observed in Kakamega, where agroforestry adoption significantly decreased BD (1.3–1.2 g cm$^{-3}$), and TN (0.2–0.4%), but increased SOC (1.0–2.0%), K (0.3–1.0 cmol kg$^{-1}$), Ca (1.1–3.0 cmol kg$^{-1}$), Cu (3.3–3.9 ppm), and Mn (23.0–29.0 ppm), while pH, Mg, S, Fe and Zn did not differ between adoption practices in Kakamega (Table 2).

Soil available S in the study did not differ between adoption practices varied significantly between the study locations (Table 2). Micronutrient contents of soils in the study differed significantly (P < 0.05) between the study locations and between adoption practices (Table 2). Agroforestry increased Cu (2.4–4.0 mg kg$^{-1}$), Mn (14.0–29 mg kg$^{-1}$) and Zn (0.9–1.6 mg kg$^{-1}$) over the control in the study. Whereas Fe did not differ between the study locations and adoption practices (Table 2).

## 3.2 Spatial distribution of soil properties as influenced by agroforestry adoption in Busia and Kakamega

### 3.2.1 Soil organic carbon and pH.
The average soil pH in Busia (5.8) and Kakamega (5.5) were within the critical range (5.5) for maize production. However, soils under non-agroforestry adoption recorded a higher pH than those under agroforestry adoption, with more than 50% of all soil samples from both locations recording a pH of > 5.5 (Fig 2). The

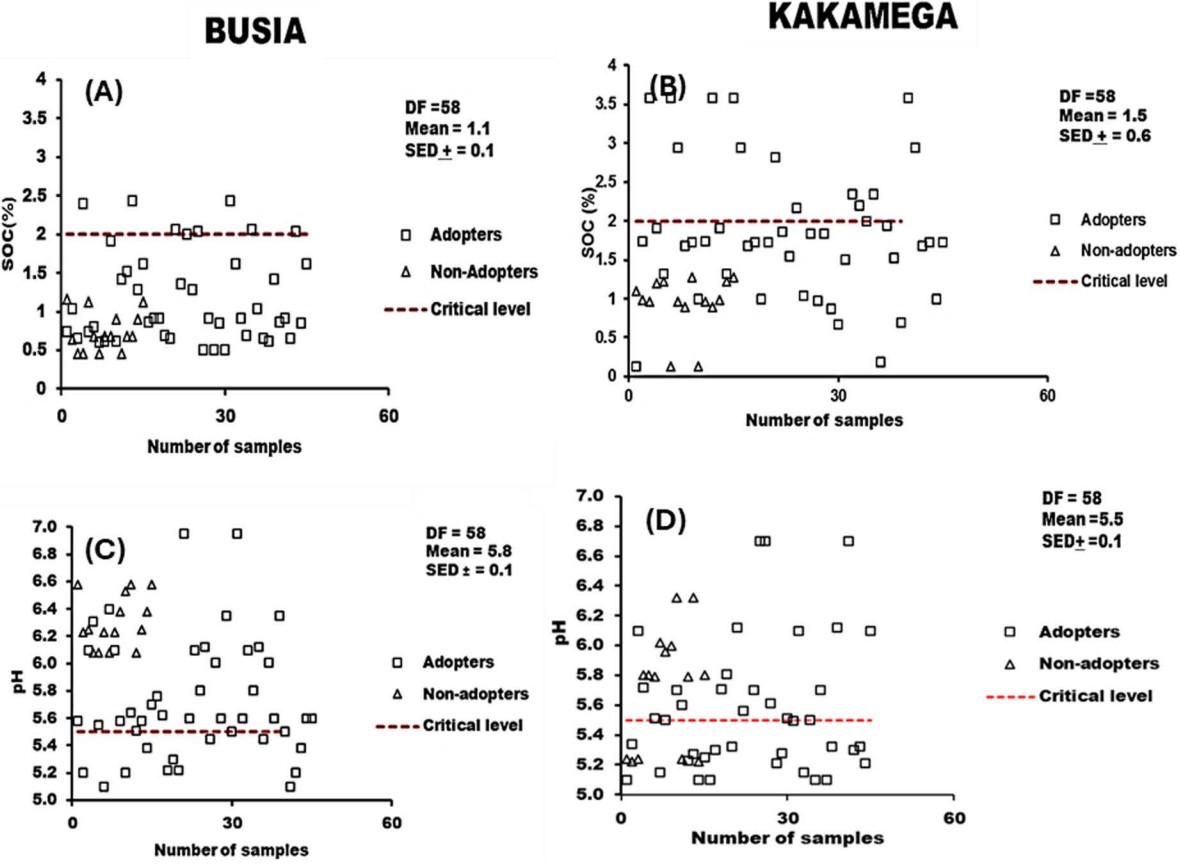

**Fig 2. Distribution of SOC (A-B) and pH (C-D) and their respective critical levels in Busia and Kakamega.**

distribution of SOC in the study revealed that about 88% of all samples (both adopters and non-adopters) recorded low SOC contents ( < 2.0%), below the critical threshold for maize production in Kenya (Fig 2).

**3.2.2 Distribution of macronutrients in soils under small holder maize production systems in Busia and Kakamega.** Distribution of soil -N showed that more than 80% of soil samples from both locations were below the critical threshold for maize production (Fig 3). Soil available P at both locations were critically low. All samples under agroforestry and non-agroforestry adoption in the study were recorded below the critical P level (<10.0 mg kg-1) for maize production (Fig 3). The mean available S in the study (0.3 mg kg$^{-1}$) was below the critical threshold (0.5 mg kg$^{-1}$) for maize (Fig 3). Available S did not differ between the study locations, however, all samples at both locations were recorded below the critical threshold (Fig 3).

**3.2.3 Exchangeable bases in soils under smallholder maize production systems in Busia and Kakamega.** It was observed that more than 70% of the soil samples from Kakamega and 93% from Busia were below the critical nutrient level for Ca (2.0 cmol kg$^{-1}$) for maize production (Fig 4). The results also showed that about 75% of soil samples from both locations were above the critical level for K (0.4 cmol kg$^{-1}$) and differed significantly between adoption practices. The increase in exchangeable Mg at both locations was influenced by agroforestry, however, all samples from both sites were below the critical threshold (1.0 cmol kg$^{-1}$) for maize production (Fig 4).

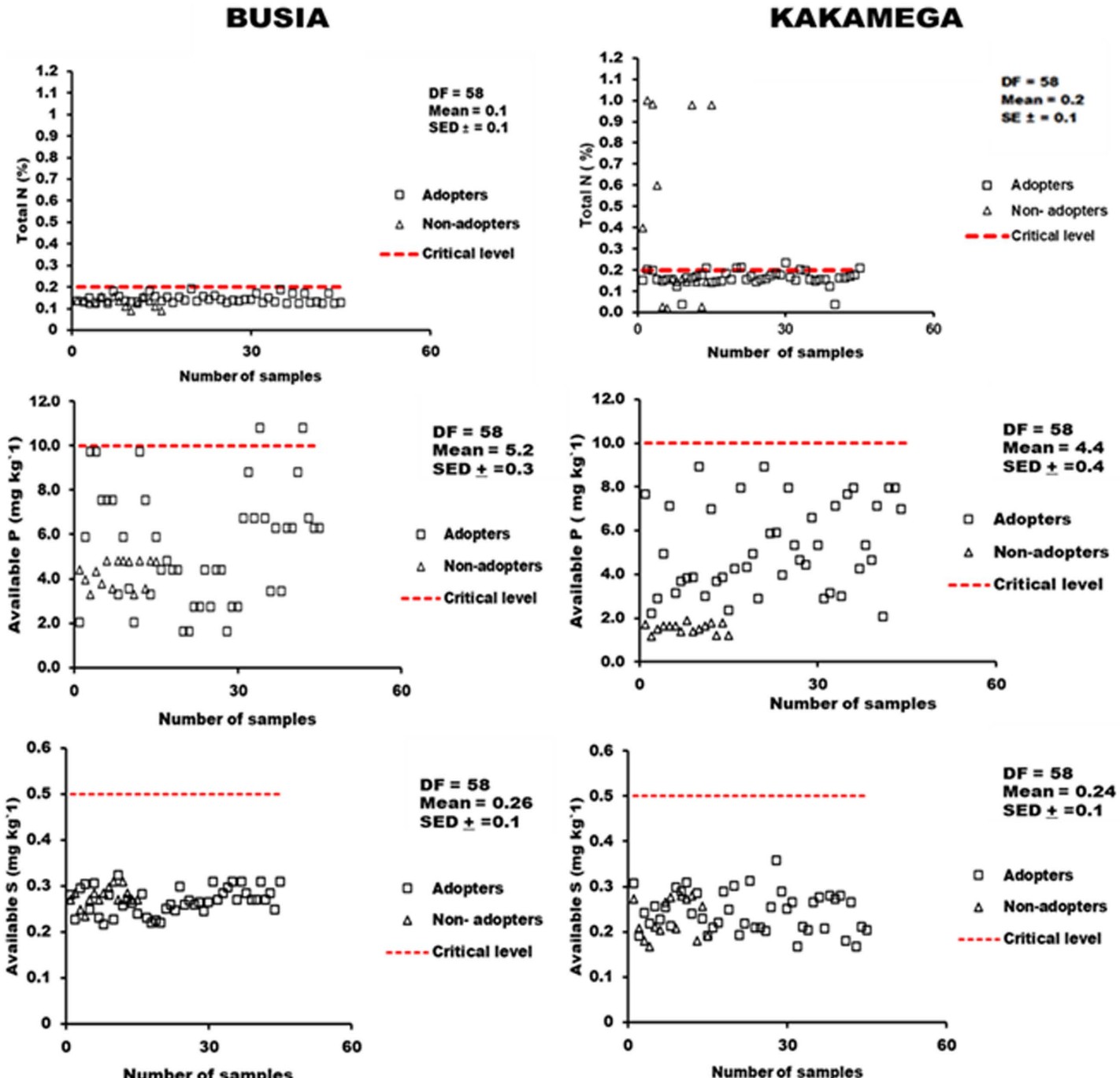

**Fig 3. Distribution of macronutrients N(A-B), P (C-D) and S(E-F) in soils under agroforestry and non-adoption in the Busia and Kakamega.**

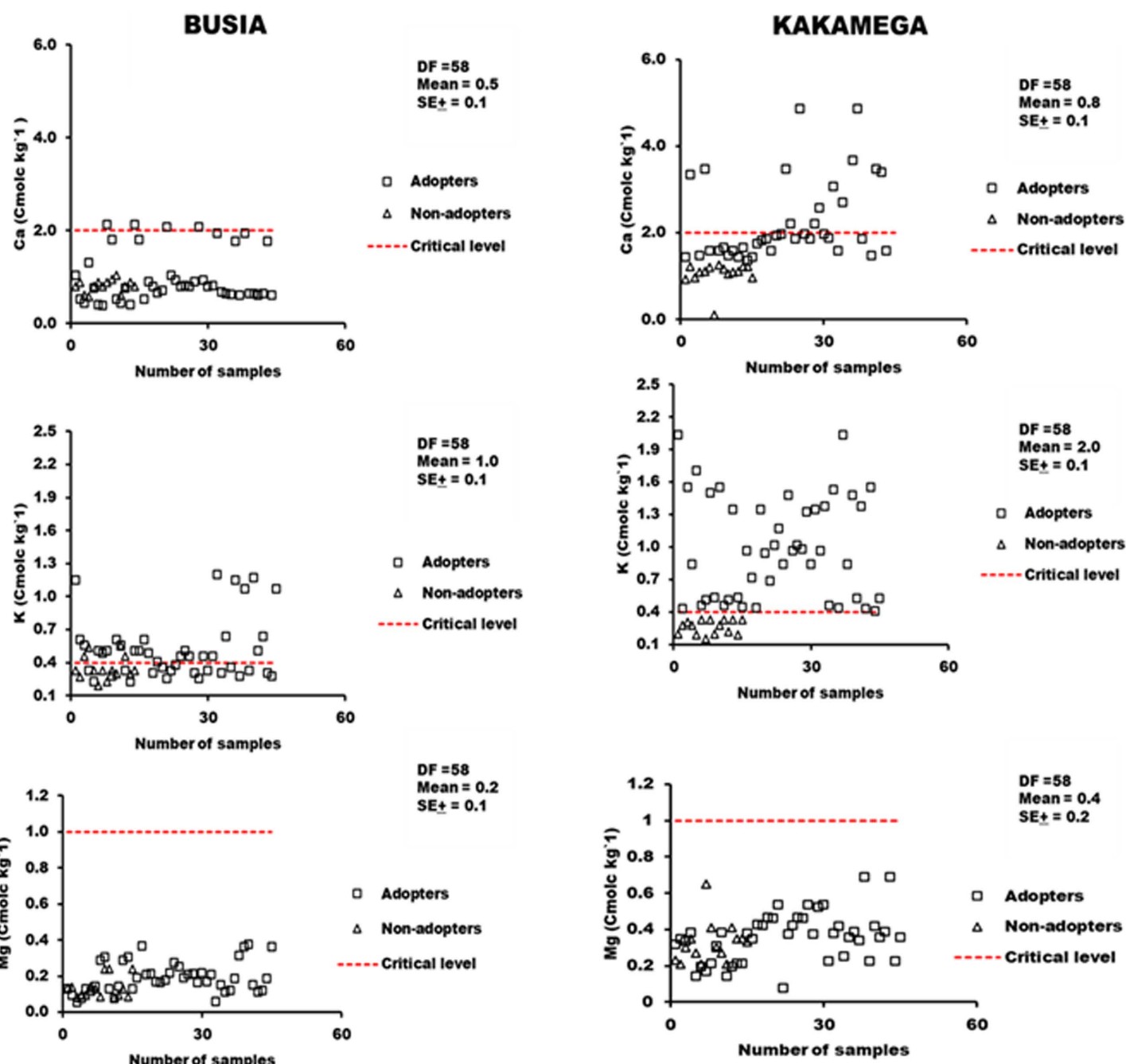

**Fig 4. Exchangeable soil bases Ca (A-B), K(C-D), and Mg(E-F) in soils under smallholder maize production in Busia and Kakamega.**

**3.2.4 Distribution of micronutrients in soils under smallholder maize production systems in Busia and Kakamega.** The availability of micronutrients in both sites and under adoption practices were above the critical level (1.0 mg kg$^{-1}$) for maize production (Fig 5). On the contrary, all samples from Busia recorded low levels of Fe, Mn and Zn contents below their respective critical levels for maize production, while only 40% of the samples from Kakamega were above the critical threshold for Mn (Fig 5).

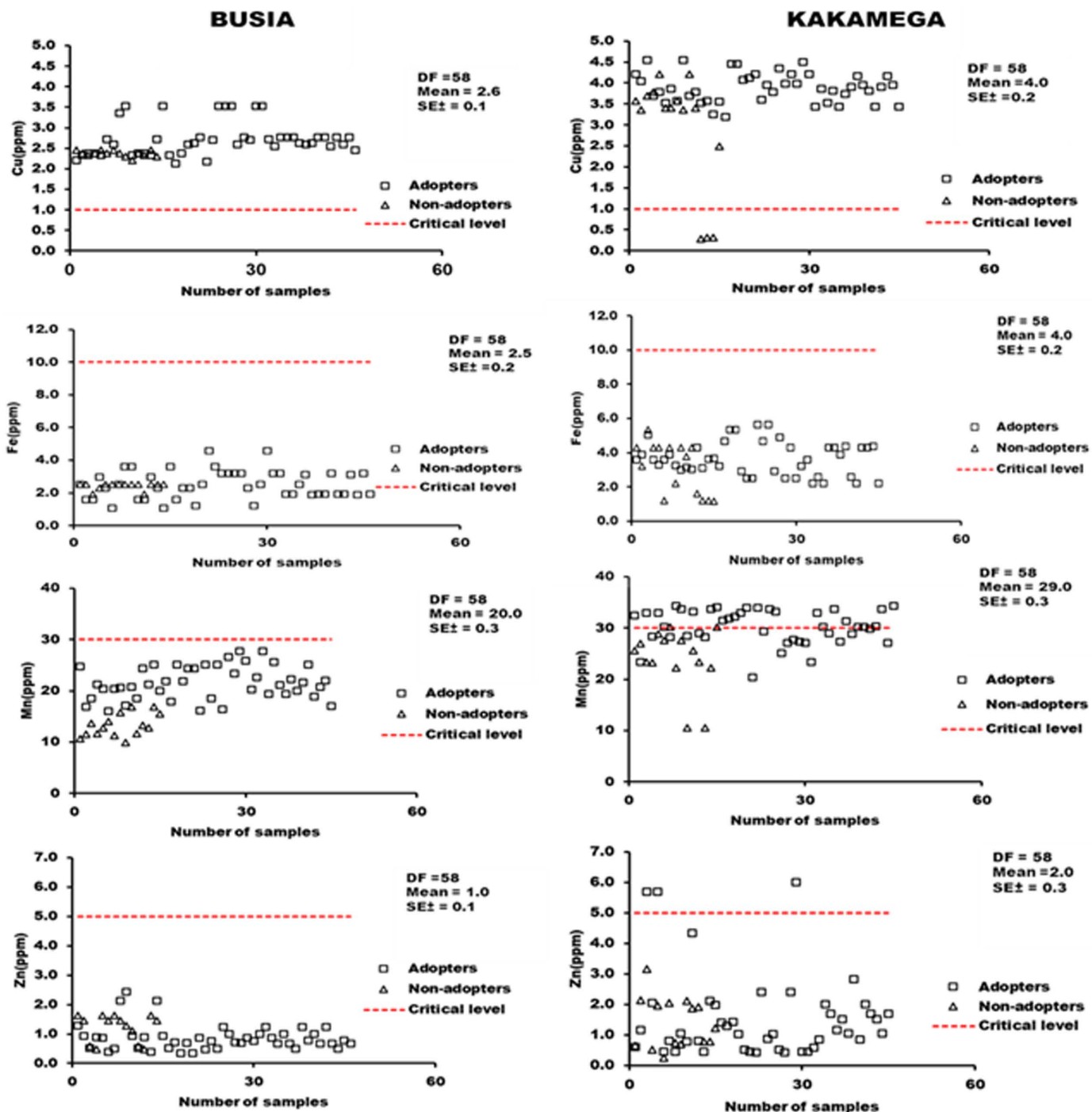

**Fig 5. Distribution of soil micronutrients Cu(A-B), Fe (B-C), Mn(E-F), and Zn(G-H) under agroforestry and non-agroforestry practices in Busia and Kakamega.**

### 3.3 Effects of agroforestry tree species on soil characteristics

Agroforestry tree species had variable effects on the physicochemical characteristics of soils at the study locations, with Leucaena and Sesbania showing the potential to enhance macronutrients and micronutrients in soils at both locations (Table 3). Significant increases in soil available P (4.3.-7.0 mg kg$^{-1}$), exchangeable K (0.4–0.7 cmol kg$^{-1}$), Mg (0.1–0.2 cmol kg$^{-1}$), and Mn (13.5–25.2 mg kg$^{-1}$) were recorded in soils under Leucaena and Sesbania compared with non-agroforestry adoption in Busia. Similar patterns were observed in Kakamega; Sesbania significantly influenced soil BD, pH, available P, and exchangeable K content of soils compared to farms under non-agroforestry adoption. Calliandra increased SOC from 0.9–2.1% over non-agroforestry in Kakamega (Table 3). Overall, it was observed that pH and K increased above their respective critical values under the canopy of Sesbania, while Leucaena increased the Mn of soils in Busia (Table 3). In Kakamega, BD, and SOC were significantly increased under the canopy of Calliandra, whereas Sesbania increased available P, S, pH, exchangeable Ca, and K. It was also observed that pH, K, and Ca in soils under the canopy of Sesbania increased above the critical values for maize production (Table 3).

## 4. Discussion

### 4.1 Influence of agroforestry Adoption soil characteristics

Changes in soil characteristics observed in the study could be attributed to variations in soil biogeochemical characteristics between the study sites and land use practices. The lower soil bulk density of 1.25 g cm$^{-3}$ recorded under agroforestry systems can be attributed to organic matter buildup as the result of litterfall and high turnover of finer roots [50]. Long-term accumulation of organic matter through litter fall has the potential to buffer soil against the impact of raindrops, compaction and erosion, hence a lower bulk density [51]. These findings agree with [52], who reported that agroforestry trees reduced soil BD from 1.9 to 1.3 g cm$^{-3}$ within a one-metre distance under the canopy of agroforestry trees in Ethiopia. Similarly, [53,54] reported improvement in soil bulk density and overall physical properties improvement of soils under agroforestry systems. Indeed, [53,55] have suggested agroforestry as an effective strategy for soil conservation due to its ability to reduce soil erosion and enhance the soil's physicochemical characteristics. Further, [55,56] observed that the addition of organic matter from decomposing leaves and wood debris is crucial to improving soil physical properties, such as BD. On the other hand, other studies such as [51,54,57] have reported significant increases in soil bulk density associated with agroforestry practices, probably due to increased compaction of soil with machinery compared to traditional tillage practices. Studies by [21,58] reported no differences in soil BD between agroforestry adoption and monocropping due to temperature extremes in arid and semi-arid conditions in SSA.

The findings of this study suggest that agroforestry did not significantly influence soil content, although the content under agroforestry adoption was higher than that under non-agroforestry adoption (Table 2). However, significant differences in BD between the study sites can be attributed to the inherent nature of the soil's physical properties, land clearing, low return of organic matter to the soil, and tillage practices observed (Table 2). The results of the current study corroborate the findings of [59] who reported no significant difference in content under agroforestry in the central highlands of Kenya.

Soil organic carbon content in the study (0.9–1.7%) was below the critical threshold (> 2.0%) for maize production in Kenya and emphasizes the need for increasing organic carbon stock at both locations. Low SOC status in the study can be caused by the low return of biomass given competing priorities for biomass such as fodder and fuel at the household level [60]. Other researchers [61,62] mentioned education, biophysical limitations, agronomic

**Table 3. Effects of selected agroforestry trees on the physicochemical properties of soils in Busia and Kakamega.**

| Location | Species | BD | SOC | pH | TN | P | K | Ca | Mg | S | Cu | Fe | Mn | Zn |
|---|---|---|---|---|---|---|---|---|---|---|---|---|---|---|
| | Units | (g cm⁻³) | (%) | – | (%) | (mg kg⁻¹) | (cmol kg⁻¹) | (cmol kg⁻¹) | (cmol kg⁻¹) | (mg kg⁻¹) | (ppm) | (ppm) | (ppm) | (ppm) |
| Busia | Calliandra | 1.3(±1.0)ab | 1.0(±1.0)ab | 5.6(±1.0)ab | 0.2(±1.0)ab | 6.1(±1.0)b | 0.5(±0.1)b | 1.0(±0.2)a | 0.2(±0.1)ab | 0.3(±0.1)a | 2.6(±0.1)a | 2.3(±0.3)a | 20.2(±0.8)b | 1.0(±0.2)ab |
| | Leucaena | 1.2(±1.0)a | 1.3(±1.0)b | 5.5(±1.0)a | 0.2(±1.0)ab | 3.3(±1.0)a | 0.4(±0.1)b | 1.0(±0.2)a | 0.2(±0.1)ab | 0.3(±0.1)a | 2.9(±0.1)a | 2.9(±0.3)a | 25.6(±0.6)c | 0.7(±0.2)a |
| | Sesbania | 1.3(±1.0)ab | 1.1(±1.0)ab | 5.9(±1.0)bc | 0.1(±1.0)ab | 7.0(±1.0)b | 0.7(±0.1)b | 1.0(±0.2)a | 0.2(±0.1)ab | 0.3(±0.1)a | 2.7(±0.1)b | 2.4(±03)a | 20.1(±0.6)b | 0.8(±0.2)ab |
| | Non-agroforestry | 1.3(±1.0)ab | 0.7(±1.0)a | 6.3(±1.0)c | 0.1(±1.0)a | 4.3(±1.0)a | 0.3(±0.1)a | 0.8(±0.2)a | 0.1(±0.1)a | 0.3(±0.1)a | 2.3(±0.1)a | 2.4(±03)a | 13.4(±0.6)a | 1.23(±0.2)b |
| Kakamega | Calliandra | 1.1(±1.0)a | 2.1(±0.2)c | 5.3(±0.1)a | 0.1(±1.0)a | 3.1(±0.2)b | 0.5(±0.1)b | 1.6(±0.2)ab | 0.3(±0.1)a | 0.2(±0.1)b | 3.8(±0.1)a | 3.6(±0.3)a | 29.0(±1.3)a | 1.9(±0.3)a |
| | Leucaena | 1.2(±1.0)a | 1.6(±0.2)bc | 5.3(±0.1)a | 0.2(±1.0)a | 5.0(±0.2)c | 1.0(±0.1)c | 1.9(±0.2)b | 0.4(±0.1)b | 0.2(±0.1)b | 3.8(±0.1)a | 3.9(±0.3)a | 31.0(±1.3)a | 1.0(±0.3)a |
| | Sesbania | 1.3(±1.0)b | 1.3(±0.2)ab | 5.8(±0.1)b | 0.2(±1.0)a | 8.0(±0.2)d | 2.0(±0.1)d | 4.2(±0.2)c | 0.4(±0.1)b | 0.3(±0.1)a | 4.0(±0.1)a | 3.4(±0.2)a | 26.3(±1.3)a | 1.7(±0.3)a |
| | Non-agroforestry | 1.3(±1.0)b | 0.9(±0.2)a | 5.7(±0.1)ab | 0.4(±1.0)b | 1.6(±0.2)a | 0.3(±0.1)a | 1.1(±0.2)a | 0.3(±0.1)a | 0.3(±0.1)a | 3.8(±0.1)a | 3.6(±0.1)a | 28.5(±0.1)a | 1.6(±0.1)a |

Means in the column followed by the same letter indicate significance at (P ≤ 0.05), according to Tukey's HSD test; SE ± = Standard Error of the mean.

practices, and limited extension services as causes of low levels of awareness of SOC loss under smallholder agricultural systems. Studies done by [63] reported increased SOC under Gliricidia-maize intercropping systems over monoculture maize due to the decomposition of organic matter, hence the need for incorporating organic matter in soils for sustainable production. Contrary to the findings of the current study, Suleiman [64] reported no change in SOC contents in soils under the influence of agroforestry due to extreme drought, which affected litterfall and biomass accumulation.

Agroforestry adoption showed a slight decrease (6.3–5.5) in soil pH in the current study, reflecting the effects of the decomposition of organic matter and the release of organic acids. This corroborates the findings by [65], who reported changes in soil pH due to the decomposition of organic matter under agroforestry systems. Soil acidity ($< 5.0$) reduces the availability of essential nutrient elements such as P, Ca and Mo due to the solubility of $Al^{+3}$ and $Fe^3$, especially under humid conditions [66,67]. Such conditions are the main factors affecting cereal yields, thus, amelioration through liming and appropriate management practices were recommended by [66,67]. However, the positive effects of organic matter decomposition provide exchange sites that bind the excess $Al^{3+}$ in acidic soils and release P, which is usually due to the persistence of $Al^{3+}$ and $Fe^{3+}$ in acidic soils in humid regions of Africa.

Significant decrease of TN contents in soils under agroforestry adoption at both study locations can be linked to the decomposition of organic matter, and the leaching of soil N due to high rainfall in the western region of Kenya [68,69]. However, a few cases of the use of calcium ammonium nitrate (CAN), as reported by farmers, could be responsible for the increase in TN among non-agroforestry adopters. Findings from the study are contrary to findings by [70,71], who observed a 9–19% increase in soil N content and the positive effects of agroforestry adoption on soil chemical fertility from (0–30 cm depth) under a soybean agroforestry system. Riyadh et al. [72] recommended agroforestry as an alternative and cheaper source of crop nutrients to sustain smallholders' productivity amidst the increasing costs of fertilizer inputs.

Available P in the study was the limiting element, with more than 50% of the samples from both sites below the critical P range (30 mg kg$^{-1}$). As in most parts of SSA, phosphorus deficiency is caused by immobilization and fixation in the presence of $Fe^{3+}$ and $Al^{3+}$ under acidic conditions [70,71]. However, an overall 45% increase in available soil P due to agroforestry in this study implied that agroforestry adoption has the potential to enhance available soil P and would require further investigation(71). Moreover, the current study confirms reports of [73], who observed an 11% increase in soil P under agroforestry practices compared with monocropping systems.

Available S in the study was below the critical threshold for maize production and can be attributed to soil type, organic carbon content, climatic, clay and N contents [74]. These findings are in accord with reports by [75], who suggested the need for including S in soil fertility management programs for smallholder farming systems in SSA where fertilizer inputs mainly focus on the primary macronutrients (N, P, and K).

Low exchangeable Ca in the study could be linked to management practices and clay mineralogy of soil in the study area [66,75]. Soil acidity and poor soil management practices are reported as a leading cause of Ca deficiency in western Kenya, hence the need for adopting sustainable management practices, such as the use of organic amendments (66,75). The findings of the study are in agreement with findings reported by [76], who reported slight changes in exchangeable Ca concentrations between agroforestry and other land use systems. Contrary to the findings of the study, [77] reported a 17–57% increase in exchangeable bases under agroforestry compared with treeless fields. Higher concentrations of Ca, especially in the non-agroforestry plots, can be attributed to the use of inorganic fertilizer and agricultural lime, whose use is determined by the

financial capability of a few farmers, thus the need for a cheaper and sustainable source of Ca. Kachaka [78] suggested agroforestry practices such as improved fallow as a sustainable means to improve soil chemical fertility and the availability of basic cations.

Agroforestry adoption increased exchangeable K above the critical threshold, which conforms with the findings of [14], who reported optimum concentrations of K in soils of western Kenya for maize. Potassium contents could be influenced by the chemistry and clay mineralogy of soils. The enhancement of soil exchangeable K in this study is an important component of regenerative agriculture and has been recommended by several researchers [79–81]. Increased exchangeable K in degraded acid soils under agroforestry is evidenced by nutrient recycling in agroecosystems after the application of post-harvest cereal residues [78]. The findings of the current study suggest that those who reported enhanced K availability in acid soils (Acrisols, Nitisols, and Ferralsols) have a poor K-selective binding capacity and are low in potassium. Exchangeable Mg in all samples was rated as low ($< 1.0$ cmol kg$^{-1}$) for maize production, according to [43]. The findings from the current study are contrary to the findings of [82], who reported a 33% decrease in exchangeable Mg under the agroforestry fallow system compared with other land use practices. Magnesium deficiency can be attributed to low organic matter content, soil acidity and crop removal of basic cations [67]. This has negative consequences on the quantity and quality of food production as Mg has control over the photosystems of crop leaves [65].

The effects of agroforestry on Cu and Mn contents in soils as reported in the current study confirm the findings of [65], who reported increased soil micronutrients under agroforestry over the control plots. In the current study, Zn and Fe were the most deficient micronutrients. The deficiency of Fe and Zn can be attributed to low SOC content and poor agronomic practices, such as low return of organic matter to the soil and low use of micronutrient fertilizer as was in the fields. Observations by [83,84] showed that precipitation and increased adsorption to reactive surfaces, such as soil organic matter and metal (hydr)oxides, affect the availability of Zn due to the mineralization of organic matter or the formation of soluble organic Zn complexes [85]. Kihara et al. [86] identified Zn as the most deficient micronutrient in arable soils across sub–Saharan Africa, affecting food and nutrition security, and recommended agronomic biofortification as a potential solution to ameliorate micronutrient deficiencies and improve crop productivity and nutritional quality of agricultural produce. Kaur et al. [80] reported a 24.8% and 50.8% increase in the total contents of Cu and Mn due to agroforestry, highlighting the role of agroforestry in nutrient cycling. These findings are in line with the works of [14], who linked high levels of micronutrient deficiency with malnutrition among children in Kenya, hence highlighting the importance of sustainable soil management practices to enhance the bioavailability of soil micronutrients [75,86].

## 4.2 Effects of agroforestry tree species on soil quality

Results of the study showed that Calliandra and Leucaena increased soil pH above the critical level (5.5) compared to Sesbania (5.31), highlighting that long-term adoption of agroforestry trees has positive effects on soil quality. Similar observations were made by [20,70], who cited the effectiveness of leguminous agroforestry trees in the alleviation of soil acidity through atmospheric N$_2$ fixation. Low N contribution from agroforestry trees could be linked to the age of the tree species, low return of organic matter, and resident time of soil organic matter as affected by environmental factors such as rainfall and temperature. The enhancement of soil available P and exchangeable bases in the study under the canopy of agroforestry trees compared to soils over non-agroforestry adoption is due to the ability of trees to recycle nutrients from deeper soil layers, and the large surface area of organic matter which facilitates cation exchange and

the adsorption of ions [87,88]. Previous studies [16, 67] highlighted the ameliorative effects of Sesbania on soil available P and subsequent crop yield. Abebe et al. [4] highlighted the crucial role of fast-growing agroforestry shrubs and trees, such as Calliandra and Leucaena, in restoring degraded soils and replenishing basic cations under intensive cultivation.

## 5. Conclusion

The findings of the study revealed that soils in Busia and Kakamega counties in western Kenya are acidic and chemically degraded as evidenced by low concentrations of SOC (< 2.0), macronutrients (N, P, S), exchangeable bases (Ca and Mg), and micronutrients (Fe and Zn) which are below the critical requirements for maize production in Kenya. Soil in Kakamega was observed to be more fertile than that in Busia, as evidenced by higher concentrations of SOC, exchangeable cations, and micronutrients. Slight changes in soil parameters such as SOC content, available P, K, Mn and Cu due to agroforestry adoption highlight the potential of agroforestry adoption to enhance nutrient availability and, hence soil quality. The effects of agroforestry adoption on nutrient availability and soil quality under smallholder maize production have not been fully explored due to the limited use of agroforestry in soil fertility programs by smallholder farmers, given competing priorities for farm resources, including fodder, biofuel, and timber at the household level. It was revealed from the study that Sesbania is more effective in enhancing the physicochemical characteristics of soils at both sites, and as such, its inclusion in soil fertility management and conservation agriculture extension messages is important. Further studies are needed to understand the mechanisms of nutrient release from different agroforestry biomass and their influence on soil characteristics and maize yield under smallholder maize production in western Kenya.

## Supporting Information

**S1 Data. Data set.**
(CSV)

## Author contributions

**Conceptualization:** Henry Tamba Nyuma.

**Formal analysis:** Henry Tamba Nyuma.

**Investigation:** Henry Tamba Nyuma.

**Project administration:** Abigael Nekesa Otinga.

**Supervision:** Ruth Njoroge, Abigael Nekesa Otinga.

**Validation:** Ruth Njoroge.

**Writing – original draft:** Henry Tamba Nyuma.

**Writing – review & editing:** Ruth Njoroge, Abigael Nekesa Otinga.

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
