## [Decision Letter · Decision Letter 0]

18 Nov 2024

PONE-D-24-48168Agroforestry adoption and its influence on soil quality under smallholder maize production systems in western KenyaPLOS ONE

Dear Dr. Nyuma,

Thank you for submitting your manuscript to PLOS ONE. After careful consideration, we feel that it has merit but does not fully meet PLOS ONE’s publication criteria as it currently stands. Therefore, we invite you to submit a revised version of the manuscript that addresses the points raised during the review process.

We look forward to receiving your revised manuscript.

Kind regards,

Timothy Omara

Academic Editor

PLOS ONE

Journal Requirements: When submitting your revision, we need you to address these additional requirements. 1. Please ensure that your manuscript meets PLOS ONE's style requirements, including those for file naming. The PLOS ONE style templates can be found at https://journals.plos.org/plosone/s/file?id=wjVg/PLOSOne_formatting_sample_main_body.pdf and https://journals.plos.org/plosone/s/file?id=ba62/PLOSOne_formatting_sample_title_authors_affiliations.pdf 2. In your Methods section, please provide additional information regarding the permits you obtained for the work. Please ensure you have included the full name of the authority that approved the field site access and, if no permits were required, a brief statement explaining why. 3. Please include a separate caption for each figure in your manuscript. 4. Please review your reference list to ensure that it is complete and correct. If you have cited papers that have been retracted, please include the rationale for doing so in the manuscript text, or remove these references and replace them with relevant current references. Any changes to the reference list should be mentioned in the rebuttal letter that accompanies your revised manuscript. If you need to cite a retracted article, indicate the article’s retracted status in the References list and also include a citation and full reference for the retraction notice.

Reviewers' comments:

Reviewer's Responses to Questions

**Comments to the Author**

1. Is the manuscript technically sound, and do the data support the conclusions?

Reviewer #1: Yes

Reviewer #2: Yes

2. Has the statistical analysis been performed appropriately and rigorously? 

Reviewer #1: Yes

Reviewer #2: I Don't Know

3. Have the authors made all data underlying the findings in their manuscript fully available?

Reviewer #1: Yes

Reviewer #2: Yes

4. Is the manuscript presented in an intelligible fashion and written in standard English?

Reviewer #1: Yes

Reviewer #2: Yes

5. Review Comments to the Author

Reviewer #1: Agroforestry adoption and its influence on soil quality under smallholder maize

production systems in western Kenya

Henry Tamba Nyuma

Comment: Include Ruth Njoroge and Abigael Otinga in the author list. As supervisors of the project, they have contributed immensely to the manuscript.

ABSTRACT

Comment: This is Not complete. You stopped at results. No discussion, conclusion and recommendation is summarized within the abstract. Please populate this abstract to make it informative.

INTRODUCTION

Expand your introduction by discussing lessons drawn globally, nationally then lastly in the study area before opening up to what you are investigating. Its improtant to open up the introduction to broaden your scope for investigation.

METHODOLOGY

Line 94: explain unique environmental characteristics that qualify the choice of the study area

State how you selected your ‘n’ and why the approach you chose is superior to other statistical approaches on population selection and choice of n

RESULTS and DISCUSSION

Results are well presented for analyzed nutrients. Results well illustrated in figures.

Interpretation well done under the discussion chapter.

Comparison of results with previous studies well done.

Citation used to standard required

Recent citations also utilized in the study

CONCLUSION

This is a well concluded manuscript. Recommendation for future studies also made.

Reviewer #2: Abstract:

Line 10: was-introduced? Why a hyphen?

Line 24-25: Can Sesbania and Leucaena influence Clay? How?

Introduction:

Line 38: 36–75 billion hectares? please check the value.

Line 49-50: showed a 30-60% decline in the yields of maize ? English may be improved here

Overall, the introduction section is written meticulously

Methods:

Line 87-88: Figure 1 should be put in bracket? Please see journal style and apply throughout the manuscript.

Line 106-118: Please mention information on soil sampling depth in this section

Line 124: correct English

Section 2.4 Data Analysis: Please explain how one way ANOVA was conducted with Paired observations? Or it was done between agroforestry tree species. Please mention clearly.

Results:

Line 180: should one put clay content as a variable to be affected by agroforestry practices? Also in Table 2? Please check throughout the manuscript and modify if required

Discussion:

Line 280: also water erosion?

Line 280-281: agree with [45] who reported….please check journal format..and change throughout the manuscript accordingly.

Line 302: can be is increased? Correct the same.

Line 309: Suleiman [60]? Refer to comment line 280-281 above.

Line 327: Riyadh et al. ([68]?

The discussion section is fairly well written and can be improved with improvement in English.

Conclusion: Some important results may be highlighted in this section, otherwise well written.

6. PLOS authors have the option to publish the peer review history of their article (what does this mean? ). If published, this will include your full peer review and any attached files.

**Do you want your identity to be public for this peer review?** For information about this choice, including consent withdrawal, please see our Privacy Policy .

Reviewer #1: **Yes: ** DR. BETTY ALOSA MULIANGA

Reviewer #2: No

---

## [Author Response · Author response to Decision Letter 1]

3 Jan 2025

Response to reviewers comments have been uploaded in the response to reviewers document.

---

## [Decision Letter · Decision Letter 1]

23 Jan 2025

PONE-D-24-48168R1Agroforestry adoption and its influence on soil quality under smallholder maize production systems in western KenyaPLOS ONE

Dear Dr. Nyuma,

Thank you for submitting your manuscript to PLOS ONE. After careful consideration, we feel that it has merit but does not fully meet PLOS ONE’s publication criteria as it currently stands. Therefore, we invite you to submit a revised version of the manuscript that addresses the points raised during the review process.

We look forward to receiving your revised manuscript.

Kind regards,

Timothy Omara

Academic Editor

PLOS ONE

Journal Requirements:

Additional Editor Comments:

Dear Authors,

I have attached some corrections to be made to your manuscript as in the attachment.

Specifically, the names of the authors should preferably be indicated in full, and figures need to be properly labelled.

Reviewers' comments:

Reviewer's Responses to Questions

**Comments to the Author**

1. If the authors have adequately addressed your comments raised in a previous round of review and you feel that this manuscript is now acceptable for publication, you may indicate that here to bypass the “Comments to the Author” section, enter your conflict of interest statement in the “Confidential to Editor” section, and submit your "Accept" recommendation.

Reviewer #1: All comments have been addressed

Reviewer #2: All comments have been addressed

2. Is the manuscript technically sound, and do the data support the conclusions?

Reviewer #1: Yes

Reviewer #2: Yes

3. Has the statistical analysis been performed appropriately and rigorously? 

Reviewer #1: Yes

Reviewer #2: Yes

4. Have the authors made all data underlying the findings in their manuscript fully available?

Reviewer #1: Yes

Reviewer #2: Yes

5. Is the manuscript presented in an intelligible fashion and written in standard English?

Reviewer #1: Yes

Reviewer #2: Yes

6. Review Comments to the Author

Reviewer #1: My comments have been adressed.

Results well presented for analyzed nutrients

Interpretation well done under discussion

Comparison of results with previous studies well done

No new comments

Reviewer #2: The authors have addressed all comments that was raised appropriately. I recomend the manuscript to be accepted.

7. PLOS authors have the option to publish the peer review history of their article (what does this mean? ). If published, this will include your full peer review and any attached files.

**Do you want your identity to be public for this peer review?** For information about this choice, including consent withdrawal, please see our Privacy Policy .

Reviewer #1: **Yes: ** BETTY ALOSA MULIANGA

Reviewer #2: No

---

## [Author Response · Author response to Decision Letter 2]

6 Feb 2025

I have uploaded all the required files, however, please ignore all duplicate files and consider the most recent uploaded, because I have made several attempts to remove the duplicate files with no success. Maybe I might some assistance to help me get rid of duplicate files through the editorial assistants.

---

## [Editor Report · Decision Letter 2]

9 Feb 2025

Agroforestry adoption and its influence on soil quality under smallholder maize production systems in western Kenya

PONE-D-24-48168R2

Dear Dr. Nyuma,

We’re pleased to inform you that your manuscript has been judged scientifically suitable for publication and will be formally accepted for publication once it meets all outstanding technical requirements.

Kind regards,

Timothy Omara

Academic Editor

PLOS ONE
---

## [Editor Report · Acceptance letter]

PONE-D-24-48168R2

PLOS ONE

Dear Dr. Nyuma,

I'm pleased to inform you that your manuscript has been deemed suitable for publication in PLOS ONE. Congratulations! Your manuscript is now being handed over to our production team.

Kind regards,

on behalf of

Dr. Timothy Omara

Academic Editor

PLOS ONE